## Research Article

psychosocial; community-based; humanitarian; refugee; migrant

**Corresponding author:**
M. C. Greene;
Email: mg4069@cumc.columbia.edu

# Mixed-methods evaluation of a group psychosocial intervention for refugee, migrant and host community women in Ecuador and Panamá: Results from the *Entre Nosotras* cluster randomized feasibility trial

M. Claire Greene[1] , Annie G. Bonz[2], Maria Cristobal[2], Alejandra Angulo[3], Andrea Armijos[4], María E. Guevara[4], Carolina Vega[4,5], Lucia Benavides[4], Christine Corrales[3], Alejandra de la Cruz[4], Maria J. Lopez[4], Arianna Moyano[4], Andrea Murcia[3], Maria J. Noboa[4], Abhimeleck Rodriguez[3], Jenifer Solis[3], Daniela Vergara[4], E. Brennan Bollman[1,6], Lena S. Andersen[5], Milton Wainberg[7,8] and Wietse A. Tol[5,9]

[1]Program on Forced Migration and Health, Heilbrunn Department of Population and Family Health, Columbia University Mailman School of Public Health, New York, NY, USA; [2]HIAS, Silver Spring, MD, USA; [3]HIAS Panamá, Panamá City, Panamá; [4]HIAS Ecuador, Quito, Ecuador; [5]Global Health Section, Department of Public Health, University of Copenhagen, Copenhagen, Denmark; [6]Department of Emergency Medicine, Columbia University Vagelos College of Physicians and Surgeons, New York, NY, USA; [7]Department of Psychiatry, Columbia University Vagelos College of Physicians and Surgeons, New York, NY, USA; [8]Department of Psychiatry, Columbia University New York State Psychiatric Institute, New York, NY, USA and [9]Athena Research Institute, Vrije Universiteit Amsterdam, Amsterdam, The Netherlands

## Abstract

Community-based psychosocial interventions are key elements of mental health and psychosocial support; yet evidence regarding their effectiveness and implementation in humanitarian settings is limited. This study aimed to assess the appropriateness, acceptability, feasibility and safety of conducting a cluster randomized trial evaluating two versions of a group psychosocial intervention. Nine community clusters in Ecuador and Panamá were randomized to receive the standard version of the *Entre Nosotras* intervention, a community-based group psychosocial intervention co-designed with community members, or an enhanced version of *Entre Nosotras* that integrated a stress management component. In a sample of 225 refugees, migrants and host community women, we found that both versions were safe, acceptable and appropriate. Training lay facilitators to deliver the intervention was feasible. Challenges included slow recruitment related to delays caused by the COVID-19 pandemic, high attrition due to population mobility and other competing priorities, and mixed psychometric performance of psychosocial outcome measures. Although the intervention appeared promising, a definitive cluster randomized comparative effectiveness trial requires further adaptations to the research protocol. Within this pilot study we identified strategies to overcome these challenges that may inform adaptations. This comparative effectiveness design may be a model for identifying effective components of psychosocial interventions.

## Impact statement

The mental health and psychosocial impacts of forced migration include both psychological and social problems. Most existing research on mental health and psychosocial interventions focuses on treating symptoms of psychological distress and common mental disorder, such as depressive, anxiety and post-traumatic stress symptoms. While psychosocial interventions that target the social dimensions of mental health and psychosocial wellbeing and focus on promoting positive mental health are available and often implemented, there is less research evidence demonstrating which intervention strategies and components are effective. This study builds on previous research where we co-designed an intervention with refugee and migrant women in Ecuador and Panamá specifically to address psychosocial problems that they had prioritized. These problems included emotional distress, sadness, xenophobia and discrimination, social isolation, loneliness, and gender-based violence. Through a community consultation process, we assembled an intervention that combined local strengths-based strategies to promote psychosocial wellbeing with evidence-based intervention components focused on psychoeducation,





problem solving and stress management. We trained pairs of refugee and migrant women from the 11 study communities, most of whom had no prior experience with mental health and psychosocial support interventions, to deliver this intervention to a total of 225 migrant and host community women. We implemented a mixed methods study of two different versions of the intervention to assess the feasibility of research and intervention protocols. Results from this study provide promising indications of the acceptability, appropriateness and safety of this intervention. Further adaptations are needed to overcome contextual challenges to implementing and evaluating this intervention in community settings, such as population mobility and barriers to attendance. Further research evaluating this intervention and using this comparative effectiveness model to test specific intervention components is needed to advance the evidence on community-based psychosocial interventions for displaced and migrant populations.

## Introduction

In 2020, 3.6% of the global population – 281 million people – had migrated and/or were residing across international borders (IOM, 2022). In recent years there has been an increase in overall migration, labor migration and forced migration, with the highest growth rate observed in Latin America and the Caribbean (IOM, 2022). The economic and political crisis in Venezuela has led to over 6 million people being displaced primarily to countries within Latin America (UNHCR, 2022). Continued conflict in Colombia and El Salvador as well as climate-related disasters in Honduras, Cuba, among other countries in the region, have also contributed to high levels of displacement and migration (IOM, 2022).

Forced migration is associated with an increased risk of mental health and psychosocial problems among asylum seekers, refugees and migrants in Latin America (H. Carroll et al., 2020; Espinel et al., 2020; Salas-Wright et al., 2022). Research supporting the effectiveness of scalable psychological interventions to reduce symptoms of common mental disorders and psychological distress, essential aspects of mental health and psychosocial well-being (Bangpan et al., 2019), in populations affected by humanitarian emergencies has increased (Barbui et al., 2020; Fuhr et al., 2020; Tol et al., 2020). However, displacement and migration also present psychosocial challenges that extend beyond symptoms of distress and common mental disorder. In preliminary research, we identified that in addition to psychological problems, social protection problems (e.g., gender-based violence, feeling rejected and discrimination) were salient aspects of psychosocial wellbeing among displaced women from Venezuela (Greene et al., 2022b).

Research on promotion and prevention interventions that emphasize the social dimensions of wellbeing is scant and often suffers from methodological limitations (Haroz et al., 2020). One limitation is the lack of evidence on which components of psychosocial interventions are effective in improving mental health and the mechanisms by which they operate. We piloted a community-based psychosocial intervention, *Entre Nosotras* ('among/between us' in Spanish), that was designed to address the psychological and social dimensions of wellbeing for women in Ecuador and Panamá (Greene et al., 2022b). In this study we tested two versions of the intervention to determine the feasibility of research and intervention protocols. A future definitive evaluation could then clarify whether specific components enhanced intervention outcomes. Specifically, we included a modified version of an evidence-based stress management intervention to one study condition to evaluate (first) the feasibility, and (in the future) potential unique effect, of this component designed specifically for adults experiencing adversity. Adding this psychological component could serve as a model for optimizing community-based psychosocial interventions in future research.

### Study objectives

The objective of this feasibility trial was to evaluate the appropriateness, acceptability, safety and feasibility of the *Entre Nosotras* intervention and research procedures to inform the design of a fully-powered cluster randomized comparative effectiveness trial.

## Methods

This study was a two-arm comparative effectiveness cluster randomized feasibility trial of two active group psychosocial interventions designed for displaced and migrant women in Ecuador and Panamá. The trial was approved by the Ethics Committees at Columbia University Irving Medical Center, Universidad de Santander and Universidad San Francisco de Quito. The trial protocol was published and registered online (NCT05130944) (Greene et al., 2022a).

### Study setting

This project was implemented in collaboration with HIAS, an international non-governmental organization that provides protection services, including community-based mental health and psychosocial support, to refugees, migrants and other vulnerable persons. The study was implemented in 11 communities nested within three sites in Ecuador (Guayaquil and Tulcán) and Panamá (Panamá City/Panamá West). Guayaquil is a large coastal city and a destination for many refugees, asylum seekers and migrants (hereafter referred to as 'migrants') from Colombia and Venezuela. Tulcán is a rural area located near the Ecuador–Colombia border and often a temporary place of transit for migrants. Panamá City is the capital of Panamá and a destination for migrants, primarily from Central and South America. Many migrants settle in surrounding peri-urban areas. The majority of migrants in Panamá and approximately half of the migrants in Ecuador are female, most of whom are of reproductive age (Blyde et al., 2020). These three sites were selected because they have large migrant populations and are diverse in terms of urbanicity, service delivery systems, populations and other implementation factors.

### Participants and procedures

#### Randomization and blinding

Within the three sites, we randomly allocated at least half of the communities to receive an enhanced version of the intervention, *Entre Nosotras*, that included an additional stress management component. Participants in the remaining communities received the standard version of *Entre Nosotras*. Communities (clusters) were randomly allocated to study conditions using a random number generator in Stata by a researcher not affiliated with the project. Two pairs of communities adjacent to each other with

overlapping catchment areas were combined into two clusters (versus four independent community clusters), thus leading to nine randomized community clusters. It was not possible to blind participants, intervention providers or outcome assessors to study conditions.

## Study conditions

*Entre Nosotras* is a community- and strengths-based intervention designed to mobilize social support, strengthen community connectedness and stimulate collective action to promote the safety and wellbeing of migrant women. The intervention was composed of existing intervention components as well as locally designed elements assembled through a community consultation process. The participatory design of the intervention involved identifying and characterizing priority psychosocial problems among migrant women in the community through qualitative methods, conducting theory of change and intervention design workshops, followed by mock sessions to pilot and iteratively refine the intervention (Greene et al., 2022b). The resulting intervention aimed to address a range of problems prioritized by community members including social problems (e.g., interpersonal violence, xenophobia and discrimination, social isolation and loneliness) and psychological problems (e.g., emotional distress and sadness).

The intervention included five weekly two-hour sessions delivered by female facilitator pairs within the community and designed to be adaptable across each study site (Supplemental Table 1). The intervention was designed using content from the HIAS Mental Health and Psychosocial Support Curriculum (HIAS, 2021), Psychological First Aid (World Health Organization, War Trauma Foundation, and World Vision International, 2011), Problem Management Plus (World Health Organization, 2016), participatory methodologies (Soliz and Maldonado, 2012) and the Community Action Cycle (Save the Children and Pathfinder, 2013). The enhanced study condition integrated a *stress management* component into each of the five sessions of the standard *Entre Nosotras* intervention. The stress management component was based on the World Health Organization's Self Help Plus intervention, and included audio exercises focused on skills for managing stress (World Health Organization, 2020, 2021). Within each session, facilitators introduced the skill(s) and the participants practiced the skill with the support of audio exercises in Spanish. The objective of the study was to test the feasibility, as opposed to the effectiveness, of the intervention and associated trial procedures. Therefore we did not include a no-treatment control group.

## Recruitment, screening and informed consent

Participants were recruited by referral from HIAS staff, community workers and community leaders, and through community outreach by research assistants. Research assistants contacted individuals by phone or in person to provide information about the study. Interested individuals were invited to complete a screening after providing verbal consent. Participants were eligible if they were 18+ years of age, identified as a woman, were currently residing in the study community, spoke and understood Spanish, and reported no to moderate psychological distress (Kessler-6 < 13) (Kessler et al., 2003, 2010). In Ecuador, both migrant and host community women were included because the integration of these communities emerged as a community- and organizational-priority during the formative research in Ecuador and was expected to promote migrants' wellbeing (Berry, 1985; Berry and Hou, 2021). In Panamá, migrant women were included in the study. Participants in both sites were excluded if they reported severe psychological distress (Kessler-6 score ≥ 13), disclosed suicidality or displayed cognitive impairment that would prevent participation in a group psychosocial intervention. Excluded participants were referred to HIAS for further assessment and services. Interested eligible participants provided written informed consent before completing a baseline assessment.

## Data collection and measures

Participants completed a baseline assessment within 1 week of enrollment and attended the first of five weekly intervention sessions within 2 weeks of completing the baseline. Two follow-up assessments were conducted by a research assistant after the intervention (i.e., 5 weeks post-enrollment) and again 5 weeks post-intervention (i.e., 10 weeks post-enrollment).

During this follow-up period we selected 10 facilitators and up to 30 participants per site to complete in-depth qualitative interviews to explore the acceptability, appropriateness, feasibility and safety of the intervention and research procedures. We randomly selected participants within the following strata: intervention completers (4–5 sessions), low attenders (0–2 sessions), participants with high baseline distress (Kessler-6 score > 10), participants with low baseline distress (Kessler-6 score < 5), and both host and migrant community members. Assessments were designed to be conducted remotely or in person, depending on COVID-19 policies and recommendations. Participants who completed the assessments in person were reimbursed for transportation costs and given a take-away snack. Participants who completed the assessments remotely were reimbursed for airtime or internet connectivity.

We collected participant-, service- and implementation-level measures to evaluate appropriateness, acceptability, safety and feasibility using qualitative and quantitative measures (Proctor et al., 2011). Participant-level outcomes assessed at baseline and both follow-up assessments aligned with the priority outcomes identified in the formative research (Greene et al., 2022a,b). Assessment tools included the Brief COPE to measure coping (Carver, 1997); the World Health Organization Disability Assessment Schedule (WHODAS) to measure functioning (World Health Organization, 2010); the Kessler-6 to measure psychological distress (Kessler et al., 2002); the Personal Wellbeing Index (PWI) to measure psychosocial wellbeing, including the subscales of community connectedness and sense of safety (International Wellbeing Group, 2013); and the Oslo Social Support Scale (OSS-3) to assess social support (Kocalevent et al., 2018). The PWI, the 10-item Kessler scale, Brief COPE and WHODAS have demonstrated adequate psychometric properties in Spanish-speaking populations (Moran et al., 2010; Terrez et al., 2011; Serrano-Duenas et al., 2020; Perez-Belmonte et al., 2021).

Service-level outcomes included participant attendance and adverse events, which were captured using structured intervention monitoring forms completed at each *Entre Nosotras* session, and perspectives on intervention appropriateness, acceptability and safety.

Implementation-level measures included a fidelity checklist that was completed by facilitator pairs at the end of each session. A member(s) of the research team also observed at least two sessions of the intervention per group and completed the fidelity checklist during these sessions to enable a comparison of the self vs. external fidelity evaluations. Facilitator competency was assessed by a research team member during observed intervention sessions using six items from the Enhancing Assessment of Common Therapeutic Factors (ENACT) rating scale (Kohrt et al., 2015a,b). We measured intervention usability from the perspective of facilitators at the end

of the final *Entre Nosotras* session using the 10-item Intervention Usability Scale (Lyon et al., 2021). We gathered other information on primary outcomes (appropriateness, acceptability, safety and feasibility) through structured intervention monitoring forms, tracking logs (e.g., recruitment rates) and qualitative interviews.

### Analyses

We examined the distribution of baseline participant-level characteristics in the overall sample and stratified by study condition and site. We calculated effect sizes to compare the magnitude of differences in these variables by site and condition using Cohen's d for continuous variables compared across study conditions, Eta-squared for continuous variables compared across site, and Cramer's V for categorical variables compared across site and study conditions.

We evaluated the psychometric performance of psychosocial outcome measures by assessing the internal consistency using Cronbach's alpha, internal construct validity using confirmatory factor analyses and external construct validity by estimating the correlation among baseline levels of psychosocial outcomes. Fit of the CFA models was evaluated using the chi-squared test statistic (model vs. saturated), Comparative Fit Index (CFI), Tucker-Lewis Index (TLI), Root Mean Square Error of Approximation (RMSEA) and Standardized Root Mean Square Residual (SRMR). We reported the mean difference and 95% confidence interval from baseline to each follow-up timepoint by study condition to explore sensitivity to change for the psychosocial outcome measures. We reported Cohen's d effect sizes for within- and between-group changes.

To evaluate service- and implementation-level outcomes, we employed a mixed-methods, explanatory analysis approach. We first described the distribution of attendance by study condition and community. We examined baseline correlates of intervention completion, defined as attending four or more sessions, as well as attrition, which was defined as missing one or both follow-up assessments. Baseline correlates included demographic, migration and psychosocial characteristics, which were regressed on binary service-level outcomes (e.g., intervention completion and attrition) using logistic regression. We reported descriptive statistics on other service- and implementation-level indicators. These included the proportion of individuals screened who were eligible, intervention fidelity, intervention usability, recruitment rate, facilitator competency levels, contamination and adverse events. We used a thematic approach to analyze the qualitative interviews. All themes emerging from the data were mapped onto larger domains that aligned with the study outcomes (appropriateness, acceptability, feasibility and safety) (Braun and Clarke, 2006; Braun and Clarke, 2014). Three coders reviewed 10% of the study transcripts to develop a preliminary codebook and achieved 98.38% agreement when applying those codes to two additional transcripts prior to independently coding the remaining transcripts. Qualitative themes and codes were used to explain quantitative findings.

### Results

### Participant-level outcomes

#### Characteristics of the sample at baseline

From the 342 women contacted from nine community clusters, 275 (80.4%) completed the screening (see Figure 1) (Eldridge et al., 2016); most were eligible and enrolled in the study (*n* = 225 of 275;

81.8%). On average, we enrolled 11.5 women per week. Randomization of five community clusters to the enhanced *Entre Nosotras* intervention and four to the standard *Entre Nosotras* intervention resulted in 121 women receiving the enhanced version and 104 women receiving the standard version. Seventy-seven participants completed in-depth qualitative interviews during the follow-up period. Thirty intervention facilitators completed in-depth qualitative interviews.

At baseline women were 36 years of age, on average (SD = 11.7; Table 1). Most had completed secondary school (52.0%) or college (24.7%). Approximately half were unemployed (53.8%) and 30.5% engaged in informal work. We observed lower levels of education and higher rates of unemployment among participants in Tulcán relative to Guayaquil and Panamá. Approximately two-thirds of women were Venezuelan (65.9%), followed by Colombian (14.8%), Ecuadorian (12.6%) and other nationalities including Cuban, Dominican, El Salvadoran, Honduran, Nicaraguan or reported multiple nationalities (6.7%). In Ecuador, 19.6% of the sample were members of the host community. In Panamá, all participants were migrants, primarily from Venezuela (45.1%), Colombia (37.8%) or Central American and Caribbean countries (17.1%). One-quarter of migrants in the sample (26.9%) had arrived within the last year. The most common reasons for leaving their country of origin included economic difficulties (43.8%), family reasons (25.8%), political violence or armed conflict (14.4%), or work and other opportunities (7.2%). Participant-level differences in demographic, migration or psychosocial characteristics at baseline were small (effect sizes < 0.5).

### Psychosocial outcomes

The reliability and validity of outcome measures displayed variable results. Internal consistency at baseline was adequate for the Personal Wellbeing Index ($\alpha = 0.781$), the Brief COPE ($\alpha = 0.784$) and the WHO Disability Assessment Schedule ($\alpha = 0.842$). The Kessler-6 ($\alpha = 0.483$; 6 items) and Oslo Social Support Scale ($\alpha = 0.539$; 3 items) displayed low internal consistency. The fifth item on the Kessler-6 ('*feeling like everything was an effort*') revealed low item-rest correlation and, if removed, would result in a five-item measure of psychological distress with improved, although still low, internal consistency ($\alpha = 0.594$). This item has displayed low loadings and weak item-rest correlations in a previous validation study of the 10-item Kessler scale in Ecuador (Larzabal-Fernandez et al., 2023).

The internal construct validity of most measures revealed adequate to good fit statistics consistent with single-factor models, with the exception of the Brief COPE, which displayed the best fit for its recommended three-factor structure. The model fit of the Personal Wellbeing Index, Brief COPE and WHO Disability Assessment Schedule improved with the inclusion of covariances between specific items as determined by the modification indices. After these modifications were made, the Kessler-6 and Personal Wellbeing Index displayed good model fit. The Brief COPE and WHODAS displayed adequate fit. The Oslo Social Support Scale did not yield fit statistics enabling evaluation of its internal construct validity. Details of model fit and specification are provided in Supplemental Table 3.

The correlations among baseline psychosocial measures revealed mixed support for their external construct validity. Psychosocial wellbeing and its subscales (community connectedness, sense of safety) as well as social support were inversely correlated with psychological distress and functional impairment, as

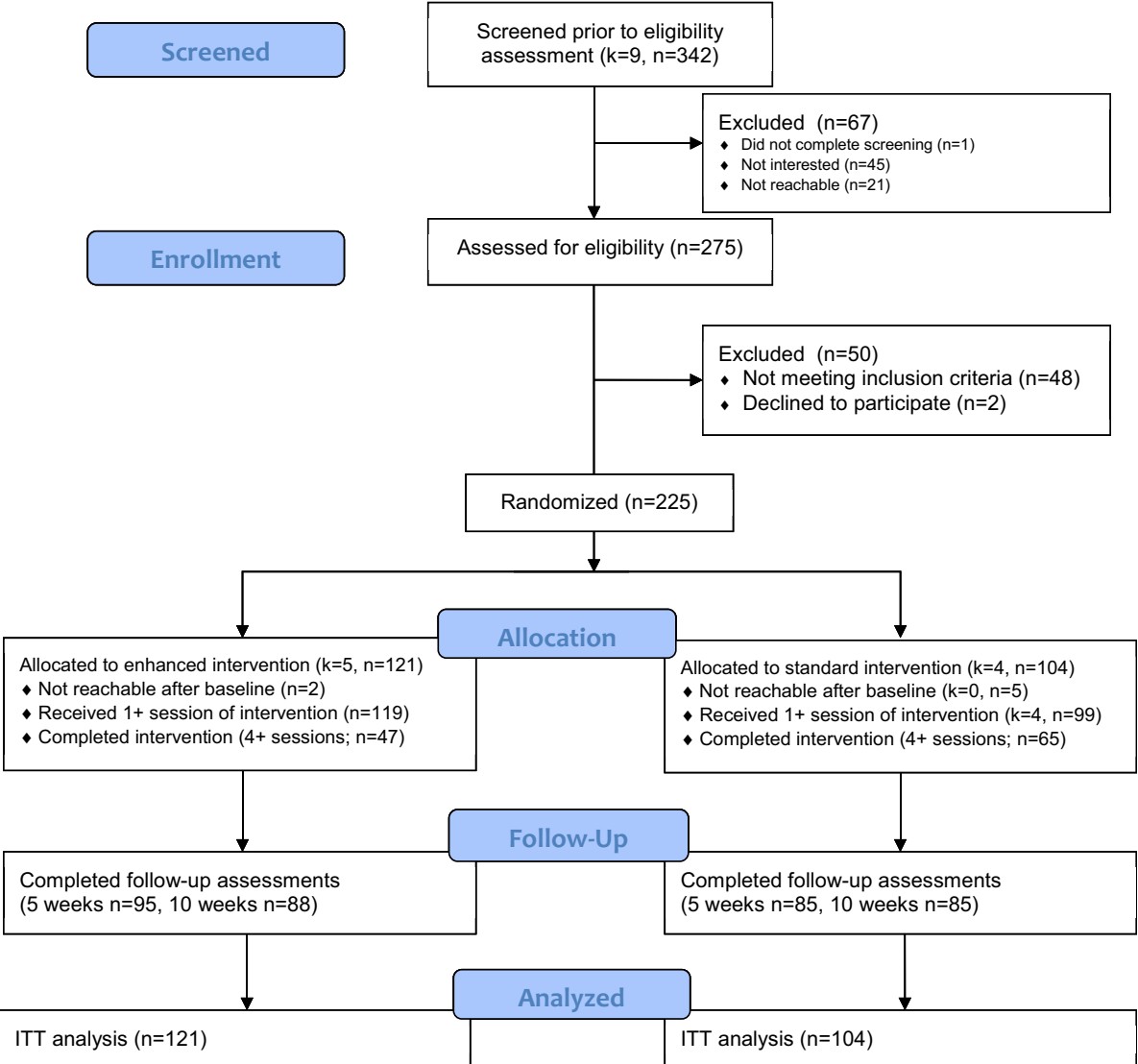

**Figure 1.** Participant flow chart.
*Note*: *k* = clusters, *n* = participants.

hypothesized. However, coping subscales revealed inconsistent and often weak relationships with the other psychosocial measures (Table 2).

We observed small to moderate within-group changes from baseline to the first follow-up assessment in psychosocial wellbeing, sense of safety and community connectedness, suggesting the appropriateness of these psychosocial outcomes ($d$ = 0.30–0.43; Table 3). The magnitude of these changes attenuated at the 10-week follow-up. These outcomes were corroborated by the changes attributed to the intervention reported in qualitative interviews. Perceived impacts included improved wellbeing and self-esteem, personal empowerment and confidence, social networks, and awareness of resources to help address safety concerns and violence.

*'What helped me the most was to manage my emotional state, that is, to give myself some time, not to let myself be trapped by the circumstances of a moment that I am going through using the tools that the (facilitators) provided us. It helped a lot to manage my stress… because at that time I was going through a bad situation. In the*

*meetings we calmed down and it was the support of the other girls that also helps a lot'.* – Participant in Guayaquil, 34 years.

These changes over time did not significantly differ between groups ($d$ = 0.00–0.27) with the exception of avoidant coping, which displayed a significantly greater decrease in the enhanced *Entre Nosotras* intervention relative to the standard intervention from baseline to the first post-intervention follow-up assessment ($d$ = 0.32). One facilitator described the contribution of the stress management component, specifically the audio exercises, present in the enhanced *Entre Nosotras* intervention:

*'The activities that I found very satisfying and beneficial for the participants were the audio exercises and introspection. Concentrating on thinking about being (present) puts you in a wave of rethinking yourself as a woman. It is like therapy. That was the part that motivated me the most and this was the (activity) that made the women look towards themselves and continue a path in which they can think differently about themselves, their daily activities, their way*

**Table 1.** Characteristics of participants at baseline

| | Full sample | Study condition | | | Site | | | |
|---|---|---|---|---|---|---|---|---|
| | (*n* = 225) | Standard (*n* = 104) | Enhanced (*n* = 121) | Effect size | Guayaquil (*n* = 72) | Panamá (*n* = 82) | Tulcán (*n* = 71) | Effect size |
| *Demographic characteristics* | | | | | | | | |
| Age (in years), *M* (SD) | 36.0 (11.7) | 37.5 (12.6) | 34.8 (10.7) | 0.24 | 34.7 (10.0) | 38.8 (11.2) | 34.2 (13.2) | 0.03 |
| Education, *n* (%) | | | | 0.14 | | | | 0.35 |
| Less than primary | 13 (5.8) | 6 (5.8) | 7 (5.9) | | 3 (4.2) | 0 (0.0) | 10 (14.3) | |
| Completed primary school | 25 (11.2) | 11 (10.6) | 14 (11.8) | | 5 (7.0) | 5 (6.1) | 15 (21.4) | |
| Completed secondary school | 116 (52.0) | 48 (46.2) | 68 (57.1) | | 32 (45.1) | 40 (48.8) | 44 (62.9) | |
| College degree | 55 (24.7) | 31 (29.8) | 24 (20.2) | | 26 (36.6) | 29 (35.4) | 0 (0.0) | |
| Other | 14 (6.3) | 8 (7.7) | 6 (5.0) | | 5 (7.0) | 8 (9.8) | 1 (1.4) | |
| Employment, *n* (%) | | | | 0.08 | | | | 0.17 |
| Unemployed or housewife | 120 (53.8) | 52 (50.0) | 68 (57.1) | | 29 (40.9) | 45 (54.9) | 47 (65.7) | |
| Informal work | 68 (30.5) | 34 (32.7) | 34 (28.6) | | 24 (33.8) | 25 (30.5) | 19 (27.1) | |
| Part-time formal work | 15 (6.7) | 7 (6.7) | 8 (6.7) | | 8 (11.3) | 6 (7.3) | 1 (1.4) | |
| Full-time formal work | 20 (9.0) | 11 (10.6) | 9 (7.6) | | 10 (14.1) | 6 (7.3) | 4 (5.7) | |
| Nationality, *n* (%) | | | | 0.21 | | | | 0.49 |
| Colombian | 33 (14.8) | 15 (14.4) | 18 (15.1) | | 0 (0.0) | 31 (37.8) | 2 (2.9) | |
| Ecuadorian | 28 (12.6) | 19 (18.3) | 9 (7.6) | | 7 (9.9) | 0 (0.0) | 21 (30.0) | |
| Venezuelan | 147 (65.9) | 67 (64.4) | 80 (67.2) | | 63 (88.7) | 37 (45.1) | 47 (67.1) | |
| Other (Cuban, Dominican, Honduran, Nicaraguan, Salvadoran, Mixed) | 15 (6.7) | 3 (2.9) | 12 (10.1) | | 1 (1.4) | 14 (17.1) | 0 (0.0) | |
| *Migration characteristics (among migrants, n = 197)* | | | | | | | | |
| How long have you lived in your current community, *n* (%) | | | | 0.17 | | | | 0.40 |
| Less than 1 year | 53 (26.9) | 18 (21.2) | 35 (31.3) | | 27 (41.5) | 10 (12.2) | 16 (32.0) | |
| 1–3 years | 89 (45.2) | 36 (42.4) | 53 (47.3) | | 35 (53.9) | 25 (30.5) | 29 (58.0) | |
| More than 3 years | 55 (27.9) | 31 (36.5) | 24 (21.4) | | 3 (4.6) | 47 (57.3) | 5 (10.0) | |
| Primary reason for moving to study community, *n* (%) | | | | 0.21 | | | | 0.36 |
| Migrated for work and/or more opportunities | 14 (7.2) | 4 (4.7) | 10 (9.2) | | 4 (6.4) | 6 (7.3) | 4.(8.2) | |
| Migrated for family reasons | 50 (25.8) | 27 (31.8) | 23 (21.1) | | 20 (31.8) | 27 (32.9) | 3 (6.1) | |
| Migrated due to political violence or armed conflict | 28 (14.4) | 11 (12.9) | 17 (15.6) | | 2 (3.2) | 22 (26.8) | 4 (8.2) | |
| Migrated due to economic problems | 85 (43.8) | 32 (37.7) | 53 (48.6) | | 30 (47.6) | 18 (22.0) | 37 (75.5) | |
| Other reasons | 17 (8.9) | 11 (12.9) | 6 (5.5) | | 7 (11.1) | 9 (11.0) | 1 (2.0) | |
| *Mental health and psychosocial outcomes at baseline* | | | | | | | | |
| Psychological distress (Kessler 6), *M* (SD) | 7.2 (3.3) | 7.2 (3.2) | 7.3 (3.4) | −0.01 | 6.9 (3.1) | 7.0 (3.4) | 7.8 (3.4) | 0.01 |
| Life satisfaction (PWI Single Item), *M* (SD) | 7.0 (2.2) | 6.9 (2.2) | 7.0 (2.2) | −0.03 | 7.7 (1.9) | 6.0 (2.2) | 7.3 (2.1) | 0.12 |
| Psychosocial wellbeing (PWI Total), *M* (SD) | 60.5 (11.8) | 59.9 (12.8) | 61.0 (10.9) | −0.10 | 63.5 (9.8) | 55.0 (12.5) | 63.8 (10.5) | 0.12 |
| Community connectedness (PWI), *M* (SD) | 7.5 (2.3) | 7.5 (2.3) | 7.5 (2.3) | 0.00 | 8.1 (1.9) | 6.9 (2.4) | 7.8 (2.4) | 0.05 |
| Sense of safety (PWI), *M* (SD) | 7.4 (2.5) | 7.2 (2.6) | 7.5 (2.4) | −0.14 | 8.1 (2.0) | 6.3 (2.6) | 7.9 (2.3) | 0.11 |
| Coping (Brief COPE), *M* (SD) | 80.4 (8.8) | 80.3 (8.7) | 80.6 (8.9) | −0.03 | 81.6 (9.8) | 80.3 (8.5) | 79.4 (8.0) | 0.01 |
| Problem-focused coping (Brief COPE), *M* (SD) | 24.3 (4.3) | 24.5 (4.2) | 24.3 (4.3) | 0.04 | 24.7 (4.6) | 24.6 (4.2) | 23.7 (4.0) | 0.01 |
| Emotion-focused coping (Brief COPE), *M* (SD) | 31.5 (5.8) | 31.5 (5.9) | 31.5 (5.6) | 0.00 | 31.4 (6.4) | 31.6 (5.9) | 31.4 (4.9) | 0.00 |

(*Continued*)

**Table 1.** (*Continued*)

| | Full sample | | Study condition | | Site | | | |
|---|---|---|---|---|---|---|---|---|
| | (*n* = 225) | Standard (*n* = 104) | Enhanced (*n* = 121) | Effect size | Guayaquil (*n* = 72) | Panamá (*n* = 82) | Tulcán (*n* = 71) | Effect size |
| Avoidant coping (Brief COPE), *M* (SD) | 15.4 (3.8) | 15.7 (3.7) | 15.2 (3.9) | 0.13 | 14.5 (4.0) | 15.9 (3.9) | 15.7 (3.4) | 0.02 |
| Social support (Oslo-3), *M* (SD) | 9.0 (2.5) | 9.1 (2.3) | 8.9 (2.6) | 0.07 | 8.9 (2.5) | 9.1 (2.3) | 9.0 (2.6) | 0.00 |
| Functional impairment (WHODAS), *M* (SD) | 22.6 (6.6) | 21.8 (6.2) | 23.2 (6.8) | −0.21 | 21.6 (7.6) | 22.2 (6.1) | 23.9 (5.8) | 0.02 |

*Note*: Effect size for continuous variables is Cohen's d for study condition (2 groups), partial eta-squared for site (3 groups), and for categorical variables is Cramer's V. We considered a small effect size to be <0.3, a medium effect size to be <0.5, and a large effect size to be ≥0.5.

**Table 2.** Correlations among psychosocial outcome measures

| | (1) | (2) | (3) | (4) | (5) | (6) | (7) | (8) | (9) | (10) |
|---|---|---|---|---|---|---|---|---|---|---|
| 1. Psychological distress (Kessler 6) | 1 | | | | | | | | | |
| 2. Life satisfaction (PWI) | −0.333 | 1 | | | | | | | | |
| 3. Psychosocial wellbeing (PWI) | −0.356 | 0.659 | 1 | | | | | | | |
| 4. Community connectedness (PWI) | −0.117 | 0.339 | 0.556 | 1 | | | | | | |
| 5. Sense of safety (PWI) | −0.250 | 0.505 | 0.738 | 0.312 | 1 | | | | | |
| 6. Problem-focused coping (Brief COPE) | −0.131 | 0.069 | 0.028 | 0.098 | −0.088 | 1 | | | | |
| 7. Emotion-focused coping (Brief COPE) | 0.073 | −0.162 | −0.110 | −0.020 | −0.217 | 0.589 | 1 | | | |
| 8. Avoidant coping (Brief COPE) | 0.328 | −0.207 | −0.223 | −0.057 | −0.230 | 0.127 | 0.313 | 1 | | |
| 9. Social support (Oslo) | −0.186 | 0.103 | 0.174 | 0.165 | 0.071 | 0.238 | 0.191 | −0.065 | 1 | |
| 10. Functional impairment (WHODAS) | 0.368 | −0.185 | −0.265 | −0.062 | −0.163 | −0.127 | 0.005 | 0.190 | −0.121 | 1 |

*of being, to continue with their things, with their problems, and with all the burdens that all women have, but to make it more bearable'. –* Facilitator in Tulcán.

### Service-level outcomes

#### Participant attendance

Seventy-eight percent of participants enrolled in the study attended at least one session and approximately half (49.8%) completed the intervention; 64.0% of participants who attended at least one session completed the intervention. The median number of sessions attended was three (IQR: 1, 5). Attendance and likelihood of intervention completion were significantly lower in the enhanced condition (Mean difference = −0.7 sessions, 95% CI: −1.2, −0.2; Completion: OR = 0.38, 95% CI: 0.22, 0.65; Supplemental Figure 1). We conducted a secondary multivariate analysis of attendance as a function of study condition and community to explore whether variation in attendance was explained to a greater extent by study condition. These results revealed that the community context largely confounded the association between study condition and attendance, which was no longer significant in adjusted models (Mean difference = −0.2 sessions, 95% CI: −1.3, 0.8). Attendance was higher in Panamá (Median = 4 sessions, IQR: 2, 5) relative to Guayaquil (Median = 3 sessions, IQR: 0, 4) and Tulcán (Median = 3 sessions, IQR: 0, 5; Supplemental Figure 2). Several participant-level characteristics were associated with the odds of completing the intervention (Supplemental Table 2). Women who were older (OR = 1.05; 95% CI: 1.02, 1.07), living in Panamá relative to Guayaquil (OR = 2.33; 95% CI: 1.22, 4.46), were members of the host population (OR = 2.85; 95% CI: 1.20, 6.79), had been living in

the community for more than 3 years (OR = 2.32; 95% CI: 1.14, 4.69), migrated for family reasons (OR = 5.50, 95% CI: 1.36, 22.22) and had lower levels of psychological distress at baseline (OR = 0.91; 95% CI: 0.84, 0.99) were significantly more likely to complete the intervention.

Qualitative interviews revealed common barriers to attendance including lack of time due to work, school, and/or family obligations; lack of resources and economic instability; fear of attending due to COVID-19 transmission risk; insecurity in the study community; or being controlled by their husbands. When sessions took place online due to COVID-19, some women did not have access to the internet, phones or technology that would enable them to attend. Most women were comfortable attending and participating in groups. However, some reported not feeling comfortable sharing their problems with others, particularly migrant women who were afraid to share their problems with members of the host community.

*'I can't take the day off (to attend the intervention) because I can't pay rent. I'm a single mother. I just live with my baby daughter and my son… HIAS does help me with the food card, but (my baby's) diapers I have to pay for. I'm very busy so sometimes I don't come (to the intervention). I start selling cold cakes and stuff at home, and I go out to sell any little thing. I can't stop working'. –* Participant in Tulcán, 25 years.

Factors that helped women overcome these barriers included motivation to attend, having financial security, living near the intervention site, being able to bring children to sessions or offering childcare, flexible scheduling tailored to participant availability, and reimbursement for transportation and/or connectivity costs.

**Table 3.** Sensitivity to change of psychosocial outcome measures

| | Enhanced *Entre Nosotras* + stress management | | Standard *Entre Nosotras* | | Between-group differences | |
|---|---|---|---|---|---|---|
| | Mean (SD) | Mean change from baseline (95% CI) | Mean (SD) | Mean change from baseline (95% CI) | Difference in mean change (95% CI) | Cohen's *d* Within/ between-group |
| *Psychosocial wellbeing (Total; PWI)* | | | | | | |
| Baseline (Week 0) | 60.99 (10.92) | – | 59.87 (12.80) | – | – | – |
| Post-intervention (Week 5) | 65.18 (9.09) | 4.71 (2.65, 6.77) | 65.14 (10.12) | 5.52 (3.36, 7.68) | −0.77 (−5.02, 3.49) | 0.43/0.00 |
| Follow-up (Week 10) | 63.77 (11.08) | 3.78 (1.72, 5.82) | 62.23 (11.14) | 3.05 (0.38, 5.72) | 0.16 (−4.67, 4.98) | 0.22/0.14 |
| *Sense of safety* | | | | | | |
| Baseline (Week 0) | 7.54 (2.38) | – | 7.20 (2.57) | – | – | – |
| Post-intervention (Week 5) | 8.00 (1.93) | 0.49 (0.00, 0.98) | 8.09 (1.59) | 0.93 (0.43, 1.42) | −0.46 (−1.22, 0.30) | 0.30/0.05 |
| Follow-up (Week 10) | 7.93 (2.12) | 0.36 (−0.18, 0.89) | 7.65 (2.06) | 0.46 (−0.10, 1.02) | −0.18 (−1.15, 0.78) | 0.18/0.14 |
| *Community connectedness* | | | | | | |
| Baseline (Week 0) | 7.53 (2.31) | – | 7.53 (2.31) | – | – | – |
| Post-intervention (Week 5) | 8.33 (1.58) | 0.78 (0.26, 1.31) | 8.31 (1.88) | 0.71 (0.24, 1.18) | 0.02 (−1.06, 1.09) | 0.38/0.01 |
| Follow-up (Week 10) | 8.14 (1.85) | 0.77 (0.23, 1.31) | 7.61 (2.06) | 0.16 (−0.37, 0.70) | 0.52 (−0.62, 1.65) | 0.16/0.27 |
| *Psychological distress (K6)* | | | | | | |
| Baseline (Week 0) | 7.26 (3.36) | – | 7.23 (3.20) | – | – | – |
| Post-intervention (Week 5) | 6.77 (3.83) | −0.43 (−1.28, 0.41) | 6.67 (3.94) | −0.38 (−1.27, 0.49) | −0.16 (−1.74, 1.42) | 0.15/0.03 |
| Follow-up (Week 10) | 6.54 (4.53) | −0.61 (−1.52, 0.32) | 6.43 (4.46) | −0.55 (−1.61, 0.51) | −0.10 (−1.85, 1.65) | 0.20/0.02 |
| *Social support (Oslo)* | | | | | | |
| Baseline (Week 0) | 8.93 (2.60) | – | 9.11 (2.34) | – | – | – |
| Post-intervention (Week 5) | 7.82 (4.57) | −1.11 (−1.96, −0.26) | 7.85 (4.26) | −1.26 (−2.10, −0.42) | 0.15 (−1.04, 1.34) | 0.33/0.01 |
| Follow-up (Week 10) | 7.02 (4.90) | −1.91 (−2.83, −0.99) | 7.86 (4.37) | −1.25 (−2.08, −0.42) | −0.52 (−2.20, 1.16) | 0.43/0.18 |
| *Functional impairment* | | | | | | |
| Baseline (Week 0) | 23.20 (6.82) | – | 21.83 (6.23) | – | – | – |
| Post-intervention (Week 5) | 22.37 (6.40) | −1.37 (−2.87, 0.13) | 21.61 (6.06) | 0.38 (−1.23, 1.99) | −1.74 (−3.91, 0.43) | 0.09/0.12 |
| Follow-up (Week 10) | 22.81 (7.11) | −0.64 (−2.31, 1.04) | 21.12 (6.27) | −0.32 (−1.70, 1.06) | −0.33 (−2.99, 2.34) | 0.09/0.25 |
| *Problem-focused coping* | | | | | | |
| Baseline (Week 0) | 24.26 (4.34) | – | 24.45 (4.21) | – | – | – |
| Post-intervention (Week 5) | 24.87 (4.33) | 0.64 (−0.45, 1.74) | 25.15 (4.61) | 0.52 (−0.62, 1.66) | 0.13 (−1.44, 1.69) | 0.15/0.06 |
| Follow-up (Week 10) | 25.07 (4.92) | 0.72 (−0.54, 1.97) | 26.08 (3.84) | 1.49 (0.47, 2.51) | −0.77 (−2.37, 0.82) | 0.28/0.23 |
| *Emotion-focused coping* | | | | | | |
| Baseline (Week 0) | 31.46 (5.64) | – | 31.47 (5.90) | – | – | – |
| Post-intervention (Week 5) | 31.85 (6.39) | 0.00 (−1.12, 1.13) | 31.39 (6.61) | −0.16 (−1.66, 1.35) | 0.16 (−1.68, 1.99) | 0.03/0.07 |
| Follow-up (Week 10) | 31.56 (6.70) | −0.57 (−1.90, 0.76) | 32.04 (5.85) | 0.37 (−0.96, 1.69) | −0.94 (−2.79, 0.92) | 0.05/0.08 |
| *Avoidant coping* | | | | | | |
| Baseline (Week 0) | 15.16 (3.88) | – | 15.66 (3.73) | – | – | – |
| Post-intervention (Week 5) | 14.58 (3.55) | −0.85 (−1.60, −0.10) | 15.73 (3.66) | 0.41 (−0.56, 1.38) | −1.26 (−2.45, −0.06) | 0.07/0.32 |
| Follow-up (Week 10) | 15.28 (3.78) | −0.10 (−0.83, 0.63) | 14.84 (3.42) | −0.60 (−1.40, 0.20) | 0.49 (−0.70, 1.69) | 0.09/0.12 |

Seventy-two percent of participants completed follow-up sessions with comparable rates of study retention observed at the post-intervention assessment (80.0%; i.e., 5 weeks post-enrollment) and the final follow-up assessment approximately 10 weeks post-enrollment (76.9%). Participants who were employed were more likely to complete all assessments (OR = 1.87; 95% CI: 1.04, 3.37). Participants who had lived in the community for over 3 years relative to less than 1 year (OR = 0.43; 95% CI: 0.20, 0.93) and those who had migrated due to violence or conflict (OR = 0.20; 95% CI: 0.05, 0.82) or

economic problems (OR = 0.26; 95% CI: 0.05, 0.82) were less likely to complete all research assessments. Participants with greater psychosocial wellbeing at baseline were more likely to complete all research assessments (OR = 1.03; 95% CI: 1.00, 1.06).

### Implementation-level outcomes

The *Entre Nosotras* intervention appeared to be safe, usable and feasibly implemented by trained, lay facilitators living in the study communities. In qualitative interviews, participants reported sufficient mitigation of potential risks, including COVID-19 precautions and carefully selecting safe locations and times for sessions. Facilitators reported an average usability score of 82.7 (SD = 10.4) out of 100, which is above the cutoff for 'acceptable' usability of 70 points (Lyon et al., 2021). Facilitators demonstrated high competency scores (Mean = 2.8 out of a maximum of 3.0; SD = 0.30). The proportion of sessions where there was >75% fidelity to intervention elements was high when assessed by an external rater (88%) and moderate when assessed by the facilitators themselves (59%; Supplemental Table 4). Implementation modalities differed by site and over time due to the COVID-19 pandemic. In Panamá and Tulcán, all sessions were delivered in person. In Guayaquil, approximately 40% of sessions were delivered online halfway through the implementation period.

We identified some challenges to the implementation of the research procedures. One pair of facilitators in each site reported applying the stress management activities in the fourth session of the standard condition (i.e., three sessions total) in their fidelity checklist. None of the external raters reported observing contamination. The study had a slower recruitment rate and higher levels of attrition than were expected with variation observed across communities and over time (i.e., different stages of the COVID-19 pandemic and rapidly changing contexts). There were no other major protocol deviations or any serious adverse events detected during the study and no major baseline imbalances by study condition despite the diverse communities included in the study.

All participant-, service- and implementation-level indicators of the primary feasibility trial outcomes (appropriateness, acceptability, feasibility, safety) are summarized in Table 4.

### Discussion

This feasibility trial aimed to examine the appropriateness, acceptability, safety and feasibility of conducting a fully powered cluster randomized trial of the *Entre Nosotras* intervention. We assessed these outcomes at the participant, service and implementation levels to determine whether the intervention and research procedures were adequate and warranted progression to a definitive trial (Proctor et al., 2011).

Overall, intervention and research procedures were considered appropriate as determined by the high proportion of those screened who were eligible, moderate sensitivity to change in some primary outcome measures that were corroborated by qualitative findings, appropriateness of the intervention as reported by participants and facilitators, and high levels of intervention fidelity as assessed by external raters. Facilitator fidelity self-ratings were lower than those reported by external raters, reflecting a more critical self-evaluation by facilitators of their own adherence to intervention procedures or bias by external raters who were research team members. Generally, intervention fidelity remained high across sites and sessions,

suggesting alignment of the content with the implementation context (C. Carroll et al., 2007; von Thiele Schwarz et al., 2019).

Facilitator reports of acceptability and usability were high. No serious adverse events or major risks of participation in the study were detected. However, participant attendance and intervention completion were low. Most participants who attended at least one session ultimately completed the intervention. Qualitative interviews suggest that the major barriers to attendance were related to contextual factors (e.g., COVID-19, limited time, competing responsibilities, distance to sites and telecommunication challenges). We identified differential rates of intervention completion across study conditions, which may be explained by differences across communities. Available data did not provide other indications of differential acceptability across study conditions, but this requires further exploration. Some factors that may have influenced differences in intervention retention across communities and sites include the variable roles and profiles of facilitators, resources across sites (e.g., transportation subsidy in Panamá), sessions offered virtually due to COVID-19, population mobility and stability across communities, employment options (e.g., informal work), and other contextual factors (e.g., climate influencing attendance). It is important to develop strategies to promote the intervention's acceptability and accessibility for participants that are tailored to each implementation context.

Adjustments to the research procedures are needed before conducting a definitive trial. The recruitment and attrition rates (e.g., attending at least one session), and performance of some outcome measures did not achieve needed target benchmarks. Indicators of instability (e.g., unemployment, recent and forced migration and higher distress) contributed to higher odds of intervention dropout and study attrition. Conducting assessments by telephone was a helpful strategy to promote retention. However, phone numbers frequently went out of service, particularly when participants migrated to other locations or were not able to purchase phone data/min. Safe strategies for maintaining connection with study participants through social media, secondary contacts or more regular interim follow-ups to track mobility or risk for attrition as well as reasons for variation in attendance across study communities should be explored in further research. The higher levels of intervention completion for participants with more stability suggest that the intervention may require further adaptation and optimization for harder-to-reach individuals whose voices may not have been as represented in this intervention design process. We identified evidence of contamination across study conditions in the fourth session as evidenced by facilitators in the standard condition reporting that they implemented activities focused on 'Acting on your values and being kind', which come from the enhanced intervention components. These instances of contamination were not observed by external raters. It is possible that this item on the fidelity checklist was interpreted as referencing other activities that were part of the standard condition. There was variation in sensitivity to change across measures suggesting that some of the selected outcomes (distress, functioning, social support, and emotion-focused and avoidant coping) need to be reconsidered. In contrast, we found that several aspects of the research design were feasible. Cluster randomization did not yield any moderate to large differences in baseline characteristics between study conditions despite the diverse communities included in the study. Also, study facilitators achieved sufficient competency levels supporting the use of a task-sharing model.

Findings should consider the following limitations. First, there was a lack of blinding in the study, which may have introduced

**Table 4.** Summary of appropriateness, acceptability, safety and feasibility indicators

| Outcome | Indicator(s) for progression to definitive trial | Means of verification | Evaluation |
|---|---|---|---|
| Appropriat-eness | *Eligibility:* >50% of persons screened are eligible. | Routine study monitoring forms | 82.5% of screened were eligible; 81.8% of screened were enrolled. |
| | *Sensitivity to change:* Small to moderate within-group changes in outcome measures ($d > 0.2$). | Brief-COPE, K-6, OSS, PWI, WHODAS | PWI (including subscales) and problem-focused coping subscale display moderate within-group changes in outcomes. |
| | *Intervention fidelity:* >75% fidelity to the intervention elements. | Fidelity assessment | 88% of externally rated sessions displayed >75% fidelity to intervention elements. 59% of self-rated sessions displayed >75% fidelity to intervention elements. |
| Accept-ability | *Intervention attendance and completion:* >85% of participants attend first session; >67% participants complete at least four sessions. | Routine study monitoring forms | 77.8% attended 1+ session, 49.8% completed 4+ sessions, but variation across communities. |
| | *Intervention usability:* Facilitators consider the intervention to have above average usability (IUS Score > 68). | Intervention Usability Scale | Average IUS total score (rescaled 0–100) was 82.7 (SD = 10.4). |
| Feasibility | *Recruitment rate:* Average rate of enrollment is five women per week, at minimum, in each site. | Routine study monitoring forms | 11.5 enrolled per week at the study level, on average. |
| | *Randomization:* No moderate to large differences in baseline characteristics between study conditions. | Demographics, Brief-COPE, K-6, OSS, PWI, WHODAS | No major imbalances. |
| | *Attrition:* >80% of participants complete baseline, post-intervention, and follow-up assessments. | Demographics, Brief-COPE, K-6, OSS, PWI, WHODAS | 71.6% completed all assessments. |
| | *Fidelity to research procedures:* No major protocol deviations. | Routine study monitoring forms | Higher numbers of persons screened than anticipated. |
| | *Facilitator competencies:* An average score of >2 on competency items, suggesting partially-fully demonstrating the competencies during intervention implementation. | ENACT | Average competency score = 2.80 (SD = 0.30). |
| | *Contamination:* Stress management components from the experimental conditions are included in all sessions for groups randomized to the enhanced intervention and none of the sessions for groups randomized to the standard intervention. | Fidelity assessment | 1 session in each site ($n$ = 3 total) reported applying the stress management activities in the 4th session of the standard condition; none of the external raters observing sessions reported contamination |
| | *Performance of outcome measures:* All outcome measures display adequate construct validity and internal consistency | Brief-COPE, K-6, OSS, PWI, WHODAS | PWI and WHODAS performed well; K-6, OSS and COPE required adaptations. |
| Safety | *Adverse events:* Detected in <10% of participants; no serious adverse events attributed to study participation. | Adverse event reporting | No serious adverse events. |

bias into the study outcome assessment. While most of the psychosocial outcomes had been previously validated in Spanish-speaking populations and/or migrant populations, none had been previously validated within the study setting and/or population. It is possible that these measures may not accurately capture the priority problems identified in the formative research and measurement error may increase the risk of bias and imprecision in results from this study. Future studies in this population may consider the modifications to the measures made in this study to improve their construct validity and reliability prior to using them as screening or outcome measures. Additionally, there was variation in some key implementation processes across sites that was determined by the implementation context and based on the suggestion of community members and other stakeholders during the intervention co-design process. For example, in Panamá, the study eligibility was restricted to migrants, and the facilitator pairs

were comprised of one non-specialist and one individual with a background in mental health. In Ecuador, participants included host community members in addition to migrants, and the facilitator pairs were comprised of two non-specialists without prior training in mental health. Differences in the COVID-19 pandemic across sites led to different implementation timelines and procedures (e.g., 40% of sessions in Guayaquil occurred online). This heterogeneity may have masked meaningful variation in some of the study outcomes. Results from this study are not able to definitively determine whether the addition of the stress management components to the standard *Entre Nosotras* intervention improved participant-, service-, or implementation-level outcomes. Few differences were observed between the two study conditions. Future directions included studies designed to evaluate the comparative effectiveness of these interventions and explore the added value of specific intervention components

through optimization trials, as well as mixed-methods analyses exploring mechanisms of change.

Notwithstanding these limitations, these findings are promising and indicate that a community-designed psychosocial intervention designed with migrant women through an extensive participatory research process and delivered through task sharing is feasible in diverse contexts. Furthermore, it was feasible to integrate locally designed intervention elements with existing evidence-based intervention components and test variations of these interventions within complex, dynamic contexts. Research procedures that balance the flexibility to work across these diverse implementation contexts while maintaining internal validity of evaluations of community-based psychosocial programs are needed to strengthen generalizable evidence on these interventions in humanitarian settings. The knowledge attained will inform further tailoring of the implementation and evaluation of the *Entre Nosotras* intervention. Additionally, further research testing specific intervention components and variations, including whether site-specific variations (e.g., inclusion of host communities and profiles of intervention facilitators) moderate intervention effectiveness and implementation outcomes, is needed to identify the active ingredients and essential elements of psychosocial interventions and to design more effective and efficient programs in community settings.

**Open peer review.** To view the open peer review materials for this article, please visit http://doi.org/10.1017/gmh.2023.37.

**Supplementary material.** The supplementary material for this article can be found at http://doi.org/10.1017/gmh.2023.37.

**Data availability statement.** The data that support the findings of this study are available from the corresponding author, M.C.G., upon reasonable request.

**Author contribution.** A.G.B., M.C.G. and W.A.T. contributed to conceptualizing the study. All authors contributed to data collection and/or analysis. M.C.G. drafted the initial manuscript. All authors provided critical revisions and approved the final manuscript for publication.

**Financial support.** This study was funded by the United States Agency for International Development (USAID) under the Health Evaluation and Applied Research Development (HEARD), Cooperative Agreement No. AID-OAA-A-17-00002. The trial sponsors had no role in data collection, management, analysis or interpretation. M.C.G. was supported by a Career Development Award and Training Grant from the National Institute of Mental Health (Grant Nos. T32MH096724 and K01MH129572).

**Competing interest.** The authors declare no competing interests.

**Ethics standard.** The trial was approved by the Institutional Review Boards at Columbia University Irving Medical Center (United States), Universidad de Santander (Panamá) and Universidad San Francisco de Quito (Ecuador). The trial protocol was published and registered online (NCT05130944) (Greene et al., 2022a).

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
