## [Reviewer Report]

March 17, 2023

Dear Editors Belkin, Bass and Chibanda,

I am pleased to submit our manuscript entitled “Mixed methods evaluation of a group psychosocial intervention for refugee, migrant, and host community women in Ecuador and Panama: Results from the Entre Nosotras cluster randomized feasibility trial” for consideration for publication in Global Mental Health. 

In this manuscript, we describe the evaluation of a flexible and adaptable community-based psychosocial intervention that was co-designed with migrant women in Latin America. The community-based psychosocial intervention, Entre Nosotras (‘among/between us’) is a five-session intervention delivered by lay facilitator pairs that integrates components of evidence-based interventions with locally developed participatory strategies to promote psychosocial wellbeing, social connectedness, and safety among displaced and host community women in Latin America. We conducted a cluster randomized comparative effectiveness feasibility trial in eleven communities within three sites in Ecuador (Guayaquil, Tulcan) and Panama (Panama City). These sites were selected to reflect diverse migration contexts and enable us to explore the role of context in the implementation and effectiveness of group psychosocial interventions. The intervention was acceptable, appropriate, safe, and feasible to implement through a task sharing model. However, we identified unique and generalizable contextual challenges across the study contexts related to attendance, participation, retention, and adaptations to promote fit to context that may inform future design and implementation of psychosocial interventions for migrant communities. 

We believe that this manuscript is appropriate for publication in Global Mental Health. This study highlights the evaluation of a novel intervention developed through a participatory co-design process and presents important implementation challenges and adaptation considerations for future psychosocial interventions.

This manuscript has not been published previously and is not under consideration elsewhere. We have no conflicts of interest to disclose. All authors have approved the manuscript for submission to Global Mental Health. The research was funded by the United States Agency for International Development (USAID) under the Health Evaluation and Applied Research Development (HEARD), Cooperative Agreement No. AID-OAA-A-17-00002. The primary author was also supported by a career development award and institutional training grant from the National Institute of Mental Health (T32MH096724; K01MH129572).

Sincerely,

M. Claire Greene, PhD MPH

Program on Forced Migration and Health

Columbia University Mailman School of Public Health

---

## [Reviewer Report]

Review of GMH-23-0058: mixed methods evaluation of a group psychosocial intervention for refugee, migrant and host community women in Ecuador and Panamá:

General evaluation’

Paper is well written with a succinct but excellent introduction. I am very happy to see that this study is both a decent evaluation of a scalable psychological intervention (drawing from WHO developed packages such as Problem Management Plus and Self-Help Plus) as well as from locally developed psychosocial interventions, which is not often done. The authors are to be lauded that they chose to do so.

The methods section is clear and, again, the authors are to be lauded that they included outcomes measures around well-being and connectedness that go beyond symptoms of mental disorder. Strong is also the use of clinical fidelity checks such ENACT.

The outcomes are well described and analyses, including issues with the psychometrics of some of the measures.

The effect size of symptom change are moderate and these are ‘within-group’ changes’. One wonders how participants would have fares without the intervention. However, there was no inactive control condition. The trial was meant to assess feasibility not effectiveness. For this a control group would be required. For a feasibility trial the criteria are met and I really liked the use of well-chosen quotes from interviews with participants and facilitators.

I hope a fully powered RCT will be possible in the future. But we need to be prepared for modest outcomes, as the stressors that these women face can only be partially be addressed by brief psychological interventions.

Major revisions

None

Minor edits

1. Line 81-844 the remark that “the psychosocial problems affecting displaced Venezuelan 83 women are not adequately captured by traditional symptom presentations of common mental disorder and psychological distress” refers to a ; ‘ study conducted by McQuaid and colleagues (2021)’. This i is actually a case study of one (1) person, so sentence need rephrasing.

2. Line 131: replace ‘and other migrants’ with ‘and migrants’ which how intergovernmental consensus does this (See (World Health Organization, 2022). And use ‘migrant and refugees ’throughout the manuscript.

3. Line 451L change ‘control condition ’to ‘standard intervention’

References

World Health Organization. (2022). World report on the health of refugees and migrants. Geneva.

---

## [Reviewer Report]

The manuscript “Mixed methods evaluation of a group psychosocial intervention for refugee, migrant, and host community women in Ecuador and Panamá: Results from the Entre Nosotras cluster randomized feasibility trial’ focuses on assessing the appropriateness, acceptability, feasibility, and safety of conducting a trial of the Entre Nosotras intervention, testing two variations of the intervention in different communities. Below, I provide suggestions to improve the manuscript, hoping that the authors find them helpful:

A key strength of this paper is that it evaluates an intervention co-designed with refugee women in Ecuador and Panamá. The co-design work is highly valuable and needed, and I commend the researchers' efforts to assess its implementation and, eventually, to scale this to an efficacy trial. Having said that, I see that the sample in this study included host community members. This made me wonder, how do you know (and provide assurances to the readers) that the intervention is relevant to a non-refugee or non-migrant population? Migrants face a particular set of psychosocial challenges, which would have played a vital role in the co-design of the intervention. The paper needs a more detailed justification of the decision to include host community members (beyond the brief description mentioned at the end of the article, in the limitations section.) It would also be essential to reflect upon and discuss what this means in terms of the psychosocial outcomes you evaluate and the finding that host community members are more likely to stay throughout the program.

METHODS

Study conditions: the study had two conditions, the standard Entre Nosotras version, and the enhanced version - There is no control intervention (defined as non-intervention). The section needs an explanation of this methodological decision and a justification for the value of the findings despite the lack of a control condition.

Data collection and measures: Had any of the instruments used in the study previously validated in Spanish? In Ecuador or Panamá? While the paper reports thoroughly on internal and external validity analysis, it is essential that the reader knows whether these measures have been previously validated. Related to this point, if the Kessler instrument showed low internal consistency and was the main screening instrument, what may this mean for the overall individual-level findings? A more thorough reflection of the implications of the study findings is needed.

ANALYSES

The second paragraph of this section starts with a sentence mentioning ‘study outcome measures.’ Please revise to psychosocial outcome measures’ to avoid confusing the readers.

The third paragraph of this section ends with a mention of ‘thematic analysis.’ Please add the source, as there are many different approaches to thematic analysis. For more details on thematic analysis’s different ‘schools,’ see Braun and Clarke’s (2022) Thematic Analysis: A practical guide (SAGE).

RESULTS

Participant psychosocial outcomes: Please provide a more detailed description of the number of follow-up measurements and the time frames. It seems like you did more than one (lines 334-336). However, in line 195, you state that follow-up assessments were conducted ‘within one week and five weeks post-intervention,’ which may lead the reader to think that you had a time range of up to 5 weeks to conduct the follow-up. Moreover, in lines 350-352, you describe changes ‘from the baseline to the 10-week follow-up, contributing to more confusion.

Service level outcomes: The high attrition rate is expected in humanitarian contexts, mainly if you work with migrant and refugee populations. However, I would like to see some proposed solutions in this paper. You are the ones who know the intervention, the people, and the nuances of the particular contexts where you deployed it. What may be happening here, and most importantly, what is the way forward regarding strategies in the future? Please provide some insights into this.

The fact that host community members and people with less distress at baseline are more likely to complete the intervention is worth noting and discussing. This comment relates to my first point about the implications of co-producing with migrants and implementing it with the broader community. It also makes me wonder whether the intervention leaves the ‘hardest to reach population’ (i.e., the most vulnerable migrants and refugees) out. What do the researchers have to say about this?

Lastly, the paper stated aim is to test two intervention variations. They find that the enhanced version has less attendance, but no elaboration on this key finding is provided. What may be the reason behind this finding?

Implementation level outcomes:

Elaborate more on the evidence of contamination commentary. Provide more detail in the text about the difference between the two intervention versions.

DISCUSSION

Overall point: A main contribution of this piece is transparency about the process of going from co-designing an intervention with the community to a fully fleshed efficacy RCT. Other researchers can learn considerably from this. Yet, this section needs a more in-depth discussion of the findings, the process, the methodological challenges, and the barriers to implementation described in the paper. What are the implications of your results for the next step of your project and the work conducted by other researchers doing similar projects? What are the lessons learned?

An example of my previous point: In line 508, the authors state that ‘several adjustments to the research procedures are needed before proceeding to a definitive trial,’ but there is no mention of what these may be. Please provide a more thorough description, you are the ones who were in the field, and I am sure you have valuable insights into the process that will contribute to your project and researchers doing similar work in other contexts.

This section also needs a more in-depth discussion of the differences between the intervention versions. This is a crucial aspect of the study. Still, there needs to be a discussion about the findings regarding differences and similarities between the two concerning the participant, service, and implementation outcomes described in the paper.

Limitations

Elaborate more on the outcomes vs. instruments hypothesis mentioned in line 519. How did you determine the psychosocial outcomes evaluated here? Did these come from the co-design exercise? If so, this should be stated in the text, and the findings discussed about the (lack of) previous validation of the instruments in the specific context/population studied.

The critical detail about recruiting host community members in some but not all sites is mentioned too late in the text (this is only described in line 522). Following up on my initial comment, the inclusion of host community members should be discussed in much more depth and earlier in the paper.

---

## [Reviewer Report]

June 7, 2023

Dear Editors Bass and Chibanda,

I am pleased to resubmit our manuscript entitled “Mixed methods evaluation of a group psychosocial intervention for refugee, migrant, and host community women in Ecuador and Panama: Results from the Entre Nosotras cluster randomized feasibility trial” for consideration for publication in Global Mental Health. 

In this manuscript, we describe the evaluation of a flexible and adaptable community-based psychosocial intervention that was co-designed with migrant women in Latin America. The community-based psychosocial intervention, Entre Nosotras (‘among/between us’) is a five-session intervention delivered by lay facilitator pairs that integrates components of evidence-based interventions with locally developed participatory strategies to promote psychosocial wellbeing, social connectedness, and safety among displaced and host community women in Latin America. We conducted a cluster randomized comparative effectiveness feasibility trial in eleven communities within three sites in Ecuador (Guayaquil, Tulcan) and Panama (Panama City). In this resubmission, we have incorporated the helpful suggestions of reviewers including clarifying some aspects of the study design in the methods, elaborating on the decision to and implications of including both migrant and host community women in two of the three study sites, more reflections and details of the measurement tools, as well as more suggestions regarding ways to overcome some of the implementation challenges identified in the study. We are appreciative to the reviewers for their insightful suggestions. 

We believe that this version of the manuscript is appropriate for publication in Global Mental Health. This study highlights the evaluation of a novel intervention developed through a participatory co-design process and presents important implementation challenges and adaptation considerations for future psychosocial interventions.

This manuscript has not been published previously and is not under consideration elsewhere. We have no conflicts of interest to disclose. All authors have approved the manuscript for submission to Global Mental Health. The research was funded by the United States Agency for International Development (USAID) under the Health Evaluation and Applied Research Development (HEARD), Cooperative Agreement No. AID-OAA-A-17-00002. The primary author was also supported by a career development award and institutional training grant from the National Institute of Mental Health (T32MH096724; K01MH129572).

Sincerely,

M. Claire Greene, PhD MPH

Program on Forced Migration and Health

Columbia University Mailman School of Public Health

---

## [Reviewer Report]

The authors have adequately addressed my concerns.

I read with great interest the excellent comments of Reviewer 2 and the rebuttal of the authors. I leave it to to Reviewer 2 to assess whether the concerns have been sufficiently addressed, but from my side I would like to remark that the inclusion of local non-migrant women in Ecuador is defendable and not unusual, even though from a research perspective this presents a complicating factor.

I was intrigued by the authors response to R2 about strategies to mitigate/prevent high attrition rates. I missed however a suggestion to work more strongly with informal and formal networks of migrants that may be formed around social initiatives, local churches, livelihoods interventions and charities.